

# The appropriation of GitHub for curation

Yu Wu[1], Na Wang[2], Jessica Kropczynski[3] and John M. Carroll[1]

[1] Information Sciences and Technology, Pennsylvania State University, University Park, PA, United States of America
[2] Samsung Research America, Mountain View, CA, United States of America
[3] School of Information Technology, University of Cincinnati, Cincinnati, OH, United States of America

## ABSTRACT

GitHub is a widely used online collaborative software development environment. In this paper, we describe curation projects as a new category of GitHub project that collects, evaluates, and preserves resources for software developers. We investigate: (1) what motivates software developers to curate resources; (2) why curation has occurred on GitHub; (3) how curated resources are used by software developers; and (4) how the GitHub platform could better support these practices. We conduct in-depth interviews with 16 software developers, each of whom hosts curation projects on GitHub. Our results suggest that the motivators that inspire software developers to curate resources on GitHub are similar to those that motivate them to participate in the development of open source projects. Convenient tools (e.g., Markdown syntax and Git version control system) and the opportunity to address professional needs of a large number of peers attract developers to engage in curation projects on GitHub. Benefits of curating on GitHub include learning opportunities, support for development work, and professional interaction. However, curation is limited by GitHub's document structure, format, and a lack of key features, such as search. In light of this, we propose design possibilities to encourage and improve appropriations of GitHub for curation.

Corresponding author
Yu Wu, yuw132@psu.edu

## INTRODUCTION

GitHub is a collaborative coding environment that employs social media features. It encourages software developers to perform collaborative software development by offering distributed version control and source code management services with social features (i.e., user profiles, comments, and broadcasting activity traces) (*Dabbish et al., 2012*). This web-based tool has attracted notable attention from both industry and academic communities. By the end of 2012, software developers hosted over 4.5 million repositories on GitHub (*Marlow, Dabbish & Herbsleb, 2013*). It has not only topped the list of preferred software hosting and collaboration services among developers (*Doll, 2013*), but also inspired a number of researchers to investigate how its features have supported software development practices (*Dabbish et al., 2012*; *Marlow, Dabbish & Herbsleb, 2013*; *Singer et al., 2013*). These prior studies have concluded that software developers make social inferences and collaborate with each other using GitHub social features (i.e., activity traces

and follow function) (*Dabbish et al., 2012*; *Marlow, Dabbish & Herbsleb, 2013*; *Singer et al., 2013*; *Wu et al., 2014*).

In addition to hosting and collaborating through the use of GitHub repositories, a new category of practice has recently emerged—software developers have begun appropriating GitHub repositories to create public resources lists (*Wu et al., 2015*). Such practices are recognized as curation—activities to select, evaluate, and organize resources for preservation and future use (*Duh et al., 2012*). In 2014 and 2015, GitHub repositories, such as awesome-python (https://github.com/vinta/awesome-python) and awesome-go (https://github.com/avelino/awesome-go), which curate resources about programming topics, gained vast popularity on GitHub. The number of curation repositories has steadily increased since then and many of them have remained among the most famous repositories on the entire platform (*Wu et al., 2014*). In light of the broad popularity of curation practices on GitHub, one might expect that motivations to participate in them and reasons that they are hosted as GitHub repositories rather than external websites are well-understood. However, research exploring the ways that social coding repository features have been appropriated for resource curation is sparse. In fact, the investigation of curation practices on social media has only recently begun and remains under-explored in general (*Duh et al., 2012*). The existing curation literature focuses on microblogging services (i.e., Twitter) (*Duh et al., 2012*; *Dabbish et al., 2012*; *Greene et al., 2011*) and media sharing service (i.e., Pinterest), leaving the nature of curation in software development practices untouched as an area of exploration.

To address this gap in the literature, we conducted semi-structured interviews with 16 GitHub curators to better understand motivations to engage in this practice. In doing so, our study aims to investigate: (1) developers' motivations that drive curation practices; (2) why GitHub is chosen for this purpose; (3) how curated resources are used; and (4) current limitations and potential future improvements for curation on GitHub. Our results suggest that curation practices on GitHub mostly grow out of software developers' internal (altruism) and extrinsic motivations (personal needs and peer recognition). Software developers choose GitHub to perform curation practices mainly because this platform provides convenient tools and attracts vast groups of people with common interests. Software developers also benefit from curation in many aspects such as better software development support, efficient learning tools, and communication with the community. Further, curation represents a case that a collaborative working space is appropriated to an end-product for communicating high quality resources, suggesting GitHub repositories can be used for communication purposes to support the larger community of software developers. However, current curation practices are restricted by document format, curation process, and are bounded by GitHub features. The addition of built-in tools, such as navigation support within curation projects and automated resources for updates and evaluation, hold potential for improving current practices. Our study contributes to a better understanding of software developers' motivation to curate resources and the nature of appropriating GitHub features for curation.

## BACKGROUND

This section reviews previous literature that explores individuals' motivation to curate, tools for curation, and current curation practice on GitHub.

### Motivations to curate in the social media era

Curation is a common practice in Archeology. It is the activity of collecting, evaluating, organizing, and preserving a set of resources for future use (*Bamforth, 1986*). In the Internet era, technology assists curation is commonly referred to by librarians and archivists as "digital curation" to preserve digital materials (*Higgins, 2011*). It can share features used for social bookmarking, where users specify keywords or tags for the Internet referencing that helps organize and share curated resources with a larger community (*Farooq et al., 2007*). There are several early popular social bookmarking tools, such as del.icio.us, which allows sharing of personal bookmarks (*Golder & Huberman, 2006*); Flickr, a photo tagging and sharing service (*Marlow et al., 2006*); and Reddit, a community-driven link sharing, comment, and rating service (*Singer et al., 2014*). Curation behaviors have been further studied since social media was appropriated to enable new forms of curation. Specifically, *Duh et al. (2012)* report the use of a third party tool, Togetter, for curating tweets, and uncover the intended purposes for these curated lists, including recording a conversation, writing a long article, or summarizing an event (*Duh et al., 2012*). *Zhong et al. (2013)* conduct surveys of Pinterest and Last.fm users and find that the majority users engage with the curation site for personal interests rather than social reasons (*Zhong et al., 2013*). A recent study examines the ways that communities leverage a variety of social tools for curation to support vital community activities in a large enterprise environment (*Matthews et al., 2014*). The authors also call for future studies on curation in public Internet communities (*Matthews et al., 2014*).

Curation on GitHub is a unique instance of curation set apart from the above studies in the following ways. First, the user body of GitHub is drastically different. Services like Twitter, Pinterest, and Reddit, are services for the general population with diverse backgrounds and interests, while GitHub is intended for a focused community of software developers. Members of the software developers' community share a set of common goals and practices, which is likely to affect their participation in curation practices as well. Second, unlike Pinterest, which itself is designed for curation of links, GitHub is an online work platform designed for software developers to collaborate with others on software projects, and curation is an appropriation of the collaborative coding features of the platform. The reasons behind such appropriation and whether GitHub features fully meet curation needs of developers are yet to be discovered. Third, the technology affordances of GitHub largely depart from the above mentioned services. Tools like Pinterest and Flickr, are designed for personal collection and sharing of hyper links. Reddit allows users to vote to promote links, but it hardly preserves resources. GitHub provides a collaborative working space, i.e., the repository, where software developers can work on the same project together and are enforced by Git workflow. Therefore, GitHub is distinct regarding user base, intended purpose, as well as technology affordances. Its appropriation for curation purpose raises an interesting question concerning user's motivations and experiences.

## Software developers' motivations for participating online communities

Researchers report two main categories of motivations that drive software developers' voluntary participation in open source software projects: (1) internal motivations, i.e., intrinsic motivations, altruism, and community identification, and (2) external rewards, including expected future rewards and personal needs (*Hars & Ou, 2001*; *Ye & Kishida, 2003*). Internal factors include ''intrinsic motivation'', which refers to the feeling of competence, satisfaction, and fulfillment as a motivator to participate in open source projects; ''altruism'' refers to software developers desire to care for others' welfare at own cost; and ''community identification'' refers to individual software developers' alignment of goals with the larger community. External factors include ''future rewards'' that are inured when software developers view their participation as investment, and expected future returns, including revenues from related products and services, human capital, self-marketing, and peer recognition; ''Personal needs'' are software developers' personal demand for their activity, for example, Perl programming language and Apache web server both grew out of software developers' self-interests to support their work (*Hars & Ou, 2001*). Both internal and external factors are important motivations that drive software developers' participation in open source projects.

The rise of social media impacts the way software developers participate in online space. Social media are often referred to as socially enabled tools, where social features are added to software engineering tools (*Storey et al., 2014*). It lowers the barrier to publishing information, allows fast diffusion, and enables communication at scale, which facilitates a ''Participatory Culture'' in the software developers' community (*Storey et al., 2014*; *Jenkins et al., 2009*). As a result, software developers increasingly participate in the community via social media that enhances learning, communication, and collaboration (*Dabbish et al., 2012*; *Doll, 2013*; *Singer et al., 2013*). Similarly, software developers are motivated to participate in order to satisfy personal needs (e.g., improve technical skills) and to gain peer recognition (e.g., recognition by the community) (*Storey et al., 2014*).

Despite the well-studied motivations for software developers' participation in online communities, software developers' motivation to engage in curation practices within GitHub by appropriating a collaboration software development features are currently under-explored.

## Prior GitHub research

GitHub has drawn attention from researchers in recent years, who have examined its features that promote transparency, such as activity traces, user profiles, issue trackers, source code hosting, and collaboration (*Storey et al., 2014*; *Dabbish et al., 2013*). Researchers have examined in detail how such transparency allows software developers to engage with software practices in the community (*Dabbish et al., 2012*; *Doll, 2013*; *Singer et al., 2013*). For example, *Dabbish et al. (2012)* find that the activity logs and user profiles on GitHub motivate members to contribute to software projects (*Dabbish et al., 2012*). *Marlow, Dabbish & Herbsleb (2013)* discover that developers use a variety of social cues available on GitHub to form impressions of others, which in turn moderates their

## Continuous Integration

*Tools for help with continuous integration*

- drone - Drone is a Continuous Integration platform built on Docker, written in Go
- goveralls - Go integration for Coveralls.io continuous code coverage tracking system.
- overalls - Multi-Package go project coverprofile for tools like goveralls

## CSS Preprocessors

*Libraries for preprocessing CSS files*

- c6 - High performance SASS compatible-implementation compiler written in Go
- gcss - Pure Go CSS Preprocessor.
- go-libsass - Go wrapper to the 100% Sass compatible libsass project.

## Data Structures

*Generic datastructures and algorithms in Go.*

- binpacker - Binary packer and unpacker helps user build custom binary stream.
- bitset - Go package implementing bitsets.
- bloom - Bloom filters implemented in Go.

**Figure 1** **A part of the README.md file of awesome-go curation project.**

collaboration (*Marlow, Dabbish & Herbsleb, 2013*). *Singer et al. (2013)* put GitHub in a larger social media environment, and learned that software developers leverage transparency of socially enabled tools across many social media services for mutual assessment (*Singer et al., 2013*).

These studies focus on how the technology accordances of GitHub and other social media affect software practices, including learning, communication, and collaboration (*Storey et al., 2014*). Curation as an emerging practice on GitHub raises interesting questions as to the reasons that such practice is thriving in the software developers' community, why it emerges and gains popularity on GitHub, and whether GitHub features fully support this type of practice.

### Appropriating GitHub for curation

Curation practices are enabled by GitHub features. Specifically, GitHub introduces a README.md file in the root directory of each repository. The contents of the README.md file are displayed in the front page of the repository, i.e., if a user visits the URL of a software repository hosted on GitHub in a browser, the README.md file will be displayed as a web page (Fig. 1) (https://github.com/avelino/awesome-go) along with repository structure and some project statistics, such as the number of forks and

stars (*McDonald & Goggins, 2013*). The content of REAME.md file can be structured with Markdown syntax (https://help.github.com/articles/basic-writing-and-formatting-syntax), which provides rich text features, including table of contents, links, tables, etc. README.md is designed for adding description and documentation for a repository (https://help.github.com/articles/create-a-repo).

Curation on GitHub appropriates the README.md file of a repository to create a list of resource indexes within one page. It categorizes resources into different themes and differentiates them into sections. Typically, each resource is recorded with the resource name and a brief description of the resource (see Fig. 1). In addition, URLs are attached to each of the curated items. Clicking a resource name (shown in blue in Fig. 1) will direct the user to the real web location of the resource.

## METHODOLOGY

To explore and understand software developers' experiences appropriating GitHub for curation, we conducted a qualitative study with 16 curation project owners. The study was approved by Penn State University Institutional Review Board, under the approval number PRAMS00044217. In this section, we describe our recruitment procedure, interview protocol, and data analysis processes.

### Participants recruitment

To identify participants engaged in curation practices, we queried the GitHub search API on 12/07/2015 using the keyword "curated list" to search for curation repositories. The query returned 896 repositories hosted on GitHub. We recorded the owner's user ID for each repository in the list, then we queried GitHub API again to fetch profiles with email addresses of each ID. The query returned 405 unique owners with email addresses, which we used to create a randomized list and sent 172 email invitations to curation project owners. Recipients were asked to engage in a semi-structured online text-based interview carried out via Facebook Messenger, Skype, or Google Hangouts. We began our recruitment process in early December 2015 and completed all interviews in late January 2016.

The resulting 16 participants included 15 males and one female with GitHub experiences ranging from six months to six years. Fourteen of the participants are professional software engineers, one was a graduate student, and one was a microbiologist. Eleven participants used the descriptive word "awesome" as the prefix to name their curation repository. The participants had a varying number of followers: five had less than 10 followers; eight had between 10 and 50 followers, and four had more than 50 followers. In the following section, we refer to individuals by participant number (from P1 to P16).

### Interview protocol

We conducted text-based online interview with participants, and each discussion lasted approximately 30 to 60 min. The interviews were semi-structured by the four general areas below.

- Motivations to curate resources,
- Reasons for technology choice (GitHub),

**Table 1  Summary of the coding scheme.**

| Theme | Category | Count ($n = 16$) |
|---|---|---|
| Curator motivations | Altruism | 10 |
| | Personal needs | 15 |
| | Peer recognition | 5 |
| Reasons for appropriating GitHub | Familiarity with GitHub | 9 |
| | Relevant context and audience | 13 |
| Usefulness of curated list | Supporting work | 7 |
| | Learning a new topic | 8 |
| | Communication | 10 |
| Limitations of curation | Immature format | 7 |
| | Hard to maintain | 5 |
| | Difficult to market | 5 |

- How curated lists are useful,
- The limitations of current curation practices (on GitHub).

Questions were administered conversationally to engage the participants, and they were open-ended enough that we could pursue new topics raised by the participant.

Participants were interviewed in English. The interview scripts were then downloaded for analysis.

## Data analysis procedure

We conducted our iterative analysis through four rounds of interviews allowing the first round of analysis to guide our second round of interviews, and following similarly in the third and the fourth. Themes and codes were identified, discussed, and refined in this process (*Lacey & Luff, 2001*).

In the first round of analysis, we performed open coding on the responses (*Strauss, 1987*), grouping examples that are conceptually similar. For each subsequent round of interview, we compared concepts and categories that are similar to previous ones. And in this process, we continued to refine our coding scheme while also revealing new ones. We discussed the codes collaboratively and repeatedly. We concluded the study after reaching the point of theoretical saturation, when categories, themes, and explanations repeated from the data (*Marshall, 1996*). A second researcher independently coded four sample interviews transcripts. Our analysis showed inter-coder agreement between the two researchers (kappa = 0.73).

In the process of coding, we recognized that some themes and categories are consistent with prior literature, i.e., curator motivations (altruism, personal needs, and peer recognition). Instead of developing new categories, we labeled them according to existing literature. The complete coding scheme is shown in Table 1.

## RESULTS

The results of our analysis describe curation practices on GitHub from the aforementioned four aspects, including (1) motivations to curate, (2) technology choice, (3) the use of

curated resources, and (4) the current limitations of curation practices on GitHub. The analysis is presented through a count of themes present in coded interviews (Table 1) and representative quotes from each of the four themes.

## Motivations to curate

Internal factors (i.e., altruism, community identification, and intrinsic motivation) and external rewards (i.e., personal needs and peer recognition) are identified as motivating factors in software developers' participation in open source software projects (*Hars & Ou, 2001*). In this study, our participants confirmed altruism (62.5%), personal needs (93.8%), and peer recognition (31.2%) motivated their participation in curation projects.

### Internal factors—altruism

Participants reported that they engaged in curation practices because other community members might benefit from their effort. For example, P3 believed the high quality of curated resources could help beginners with programming:

> "I see so many people when they take introductory classes in programming, they come to GitHub to get ready repositories...and that is overwhelming at first...so to get the started and motivated with programming I thought of collecting resources together in (P3's curation project)"—P3

P6 wanted to help people who were in a similar position to himself:

> "I'm a kind of remote engineer, then I want to create a list for someone tend to like me about product manager list, I just want to save some links for my learning purpose ... then public for someone if they're in need"—P6

### External rewards

Personal needs and peer recognition form software developers' external rewards derived from participating in open source projects (*Ye & Kishida, 2003*). These rewards also drive engagement in curation practices.

**Personal needs** were the most discussed reason for participation in curation (93.8%). Specifically, software developers reported that curation repositories improved productivity and enabled communication with others. Before creating curation projects, half of participants who were familiar with a particular set of resources relied on search engines whenever they tried to locate the URL of the resource. One important reason they chose to curate resources was to avoid such repetitive search efforts.

> "Before making the repo I had to do research each time I needed a (P12's curation topic). Now that I have a list, I just refer back to it when needed."—P12

> "I simply created my own list of the sites I found to be good. The idea really was to get out there scout for sites once and then be able to come back to a list without worrying about it having sites I found bad."—P9

In addition, a curated repository has a permanent URL, which was reported as a convenient way to share resources with others who were outside of curation repositories.

Participants stated that with curation projects, they only needed to point others to the URLs of their curation repositories. It was both convenient for them to share and for others to find. For example, P14 created the curated list so that she could conveniently share curated resources with others.

**Peer Recognition** surfaces as another important motivation for software developers' participation in curation practices on GitHub.

The community of software developers on GitHub adopts a particular way to endorse curation projects. A highly reputable software developer on GitHub, Sindre Sorhus (9.2K+ followers), creates the "awesome" (repository name) project on 07/11/2014, which is a meta list of curated lists (https://github.com/sindresorhus/awesome). It contains a community drafted "awesome manifesto" (https://github.com/sindresorhus/awesome/issues/207), which depicts guidelines and standards for curation practices, and requires that curation repositories conform to it if they want to be included in this meta list. The project currently has around 2,500 watchers, more than 35,000 stars, and approximately 4,000 forks, ranking the 2nd most starred repository created after 01/01/2014 (https://github.com/search?utf8=%E2%9C%93&q=created%3A%3E2014-01-01+stars%3A%3E1&type=Repositories&ref=earchresults).

11 out of 16 participants used "awesome" as a prefix to their curation project name in an effort to conform to the naming convention as well as to indicate the quality of the content. 10 of them mentioned that they were inspired by the original "awesome" project. 4 of them hoped to get their curation repository indexed by it, and one participant's curation project was already included in the "awesome" list, who felt a great honor (P10). P12 reported putting effort forward to improve his curated list to conform to the guidelines and standards as defined by the "awesome" project, stating "...with the Awesome endorsement I'm hoping it becomes a collection people trust" (P12). It demonstrated that our participants were putting efforts to align their goals with the larger community, i.e., conforming to the community standard for curating high quality resources, and would like to be recognized by the community.

In addition, P14 reported that her involvement in curation efforts helped her obtain her current job, and P10 reported that a company approached him and wanted to collaborate with him on his curated content. These rewards emerged as side-effects of curation efforts, not one of the guiding motivations for software developers to begin a curation project.

## The technology choice—why GitHub

Compared to the GitHub platform, the features on sites like Wikipedia might be considered better suited for hosting curation projects by providing convenient editing and collaborating features. However, popularly used and referenced curation projects for software developers are predominantly hosted on GitHub, whose features and information structures are designed for source code hosting and project collaborating (*Marlow, Dabbish & Herbsleb, 2013*; *Storey et al., 2014*) rather than creating and preserving lists of resources. In this section, we will address why curation has emerged as a common way to appropriate GitHub repositories.

### Familiarity with GitHub

Participants reported that software developers' existing knowledge about GitHub and its features, i.e., their strong media literacy (*Storey et al., 2014*) with GitHub, prompted them to choose this platform to host curation projects.

In general, software developers are familiar with GitHub's text editing format (i.e., Markdown syntax) and are comfortable using it. P4 and P7 both claimed that "GitHub was a tool that I was familiar with" and "so yes github would be a more natural tool to use." Specifically, software developers are accustomed to writing and formatting text contents with Markdown syntax. For example, P11 expressed that "...I love write in markdown format!", and P5 considered that "Github has a really easy way to write content in rich format (using Markdown) and view it."

> "...as developer, I think github is the best place for developer to collaborate with other to build good resource."—P15

Intimate knowledge about GitHub collaborating features is another factor:

> "Github is a really good platform to collaborate. Anyone could come, fork it, extend it and ask me to 'Merge' it (update my list)."—P5

> "...the advantage of using Github is other people can contribute easily."—P4

### Relevant content and potential audience

Participants also chose GitHub for curation because: (1) the curated contents were relevant within the GitHub context, (2) there was a large potential audience on GitHub, and (3) the GitHub community encouraged contributions.

A total of 15 out of 16 participants' curation projects were related to software development practices. They considered GitHub just suitable as a platform for sharing software development related contents: "...(it is) the place to be for projects like this" (P2).

GitHub has attracted a large base of like-minded users when it comes to software development, which increases the chances of matching resources with an interested audience:

> "GitHub has a very large audience/devs actively spending time in it, so it's definitely the right place to publish a project such as this..."—P1

In addition, hosting curation projects on GitHub encourages contributions. GitHub has many collaborative features. It is a common practice on GitHub for users to contribute to other projects (*Dabbish et al., 2012*; *Marlow, Dabbish & Herbsleb, 2013*; *Wu et al., 2014*). P12 reported that "GitHub can target at the right audience, and contributing is encouraged more...". P8 claimed that "...enable other people to (freely) contribute to it is very important to me (and I think other curators also feel the same) so a Git hosting site is ideal."

Participants also believed that other people on GitHub, who had more experiences and knowledge than themselves and other could enhance the repositories by contributing to lesser developed parts of the repository.

> "The main reason is collaboration...I may have some resources but other people may have even better stuffs or ideas to share."—P3

### The use of curated resources

Curated resources are useful for software developers to support their work, learn about a new topic, as well as communicate with others.

### *Supporting work*

Software developers rely on others' work to accomplish their own projects. Participants reported that they used different curated lists, including their own, as bookmarks or references to quickly locate the resources they need.

> "Before making the repo I had to do research each time I needed a (resources). Now that I have a list. I just refer back to it when needed. It serves as a good toolkit for future projects."—P12

> "I recently have created a Python repository and since I was not used with Python at all, I used awesome-python to know some libraries recommended by the community."—P11

In addition to supporting their work, participants also used the curation repository to keep track of high-quality resources in case they need them in the future. For example,

> "If I used it or I'm planning to use it, I'll add it there. If the resource is well written with tests and should be considered while selecting specific category, I'll add it too... but also I add (resources) that I checked already and found it interesting for the future projects."—P16

### *Learning a new topic*

When first encountering new development related topics, software developers often find themselves feeling overwhelmed. The complex information scope in the software developers' community makes it hard for developers to start tasks quickly. For example, P6 reported that "when we start to learn new thing, there are many things, we cannot know what should to spend time on."

A curated list that provides centralized peer-reviewed resources about a specific topic provides a starting point where developers know that they can find high-quality resources and begin learning the subject.

> "...I'm an iOS engineer. But someday I like to learn Ruby, I just go to awesome-ruby and pick some recourse for beginner. Googling is not going to help us like that."—P6

> "So say if I starting to learn a new tool and need to get started quickly. I might go to the main awesome list and search for it."—P5

### *Communication*

Communication is essential among software developers in order to transfer knowledge between stakeholders, as well as facilitate learning, coordination, and collaboration (*Storey et al., 2014*). Curation serves important communication purposes, including reduction of communication costs and creating a shared knowledge space.

> "I'm relative active in the meetup community in (P14's location). Talking to people, there is always a lot of talk about what makes a good (P14's curation topic). I created

list so that I can point to other easily... I refer a lot of people to the list who are looking at improving their (P14's curation topic)."—P14

One can share the content of resources with others in the current time as well, as reported by P7:

"...I sometimes encounter people who've watched (P7's curation repository) and didn't really like them, but my hunch is they haven't seen the great ones, so I send them to check out my list to see if I can convince them otherwise..."—P7

## Limitations of using github for curation

In this section, we summarize constraints that our participants have encountered when carrying out curation on GitHub.

### *Immature structure and format of current curated lists*

The README.md file on GitHub typically includes an introduction to each project and current curation practices mainly rely on that single README.md file to list all curated resources. Sometimes a list may grow excessively long. Participants complained that "resources are not searchable (when on a list)" (P4), and it was cumbersome for them to navigate through a long list:

"The only thing sometimes that nags me is that some of them are very long, which in some sense defeats the purpose."—P5

In a case where a curated list was too long, P6 created a shorter version of the same topic by selecting resources most important to him:

"there is another remote list...lot of stars, around 5k or more, but I find it that there are lots of resources, then when I look into, I'm scared of. Then I want to create my own list, just something I think useful for most."—P6

Another issue raised is that the brief description of each item in the curated list (noted in Fig.1) can be incorrect, inaccurate, or misleading:

"Bad description doesn't allow finding the required resource."—P16

Further, although these curated lists are intended to be collaborative efforts (i.e., multiple people suggest adding, deleting, or updating entries), there is no intuitive way for an audience to express their opinions or raise uncertainties about resources, only modify content. One participant suggested including a rating system in the curated list to help audience filter resources:

"...maybe it would be better we could Like/Dislike the resources ...sometimes the resources are sorted by name when popularity would be a better option... something like this would give us an overview of how much important some entry in a list is for the community."—P11

### *Excessive efforts to filter and maintain resources*

It requires a lot of time and efforts to navigate in this complex information space where curation takes place and to filter a handful of good resources. P14 emphasized the time constraints for curation:

> "Time. Time is hard...Digging through all of these resources takes time, and I'm usually pretty time constrained." (P14).

Due to the fast changing nature of the software industry, old resources become outdated and new resources emerge instantly. Curation repositories require efforts to be simply maintained, including getting rid of the outdated resources, and adding state-of-the-art resources. P16 reported that one drawback of the current curation practices was that curated resources have "no quality update."

### *Difficulties for marketing*

Although GitHub contains a vast and relevant user base, it does not provide mechanisms for a repository owner to distribute the list directly to the relevant audience. Our participants expressed that it was hard for them to target their repositories to users who were interested in the curation topic. For example, P10 conveyed his desire to recruit more contributors:

> "the only drawback is the lack of pull requests. I want more... (I want to) discover datasets I missed."—P10

And P4 found that it was demanding to reach out to both potential collaborators and consumers:

> "While it's easy to host a project on github, you still need to put effort into marketing it, so you get other people contributing or finding it."—P4

Unlike social media services such as Facebook, which automatically curates and recommends personalized content for each user, GitHub only contains technical features to allow users to search for information. If GitHub users are not aware of the existence of such curation projects, it would be difficult to find these resources in the first place. Therefore, admitting curation projects are embedded in the context of an abundant potential audience, they still lack mechanisms and features for marketing to parties of interest.

## DISCUSSION

Our study provides an in-depth view of curation practices on GitHub. We first assessed curator motivations in participating in curating activity, and compared them with motivations in open source participation literature. Then, we analyzed the reasons that GitHub was chosen for curation purposes. Next, we evaluated the implications of curation repositories to software developers' community. And finally, we uncovered the current limitations of curation practices. In this section, we generalize these main findings and discuss design suggestions with the hope to improve curation practice in the future.

## Curator motivations

One primary motivation for engaging in curation is altruism, which is also widely recognized as an important motivation for software developers' participation in open source projects (*Hars & Ou, 2001*; *Lakhani & Wolf, 2003*; *Ye & Kishida, 2003*). This finding indicates that helping behaviors may be a common reason why software developers are motivated to participate in some online activities. Thus, when designing systems for facilitating software development related practices, we should consider software developers' desire to support each other.

Enjoyment-based intrinsic motivations are the primary reason that software developers take part in open source projects, but were not specifically mentioned by the participants as a drive for engaging in curation. Researchers have learned that intrinsic motivations drive software developers to spend more time and effort on open source projects (*Lakhani & Wolf, 2003*), and it is positively reinforced by community recognition (*Ye & Kishida, 2003*). Therefore, many software developers commit themselves to open source projects for a relatively long time. At the same time, altruism alone is recognized as an unsustainable incentive for open source participation (*Ye & Kishida, 2003*). The comparison of motivations to curation with participation in open source projects leads to questions concerning curators' long-term engagement, such as whether intrinsic motivations are involved in driving curators, whether altruism can sustain curator's long-term participation in curation activities, and if not, whether there is a mechanism that regularly feeds curators' motivations to curate. The answers to these questions are beyond the scope of this study and requires further investigation.

## Leveraging github as a tool for communication

While GitHub is known as a tool for software projects hosting and collaborating (*Dabbish et al., 2012*; *Marlow, Dabbish & Herbsleb, 2013*), and communicating knowledge in software artifacts (*Storey et al., 2014*). This study finds that GitHub is also a good tool for communication of socially generated resources for developers. Here, we describe the features that have supported this practice.

First, GitHub features allow curation repositories to be shared easily. With a robust version control system as we as uniquely assigned public URL for each GitHub repository, GitHub guarantees the integrity and durability of curated contents and enables easy sharing. These technical features make GitHub repositories ideal for communicating resources with others.

Second, GitHub attracts potential audiences who can contribute to curation repositories. By connecting to relevant audience group, GitHub allows others to suggest potential curated items and evaluate existing ones. As such, the emerging of curation repositories indicates that software developers' community starts to utilize GitHub for communicating knowledge that is socially generated and maintained (*Storey et al., 2014*).

GitHub as a communication tool establishes its flexibility and reconfigurability. With a simple appropriation of its features, GitHub becomes a favorite tool intended for curation purposes in software developers' community. Such reconfigurability will lead to other practices besides curation, which can further benefit software developers' community. For

example, GitHub users started to appropriate GitHub repositories to write and publish software development related books (https://github.com/getify/Functional-Light-JS/), which accepted community suggestions as well as changes. Also, software developers initiated sharing training materials for others to discuss related matters as well as retrieving improvements (https://github.com/kentcdodds/es6-workshop).

## Curation to strengthen software developers' community

Onboarding new members and educating existing members are essential for communities of practice to sustain and grow (*Lave & Wenger, 1991*; *Wenger, 1998*; *Wenger & Snyder, 2000*). The results of this study demonstrate that curation repositories on GitHub reflect these core utilities of communities of practices. Software development related resources are changing rapidly, and software developers usually rely on a number of services and channels, such as Stack Overflow and Twitter, to keep themselves up-to-date with the trend (*Storey et al., 2014*). By centralizing peer-reviewed resources in an active community of relevant audience, curation repositories create a reliable channel that simplifies the process of discovering high quality resources. They are likely to reduce the amount of efforts individual member of the community spent on locating and filtering the resources. In addition, as the curated resources are peer-reviewed, they are more likely to guide one's learning of a certain topic than random resources encountered on the Internet. Thus curation repositories optimize the way that resources are disseminated and consumed in software developers' community, which in turn helps the community to grow.

The importance of curation for the software developers' community also raises interesting questions concerning the professional trajectories of the curators in the community. Curators relate the resource providers, i.e., software developers who develop tools, packages, frameworks, etc., with resource consumers, i.e., software developers who need to learn or work with tools, packages, frameworks, etc. In this sense, they are similar to brokers, who bridge different groups and control information flow in a community, and often bargain for better terms because of their unique positions (*Burt, 2004*; *Burt, 2005*; *Van Liere, 2010*). However, the existing studies of brokers either happen in the cooperate environment (*Burt, 2004*; *Burt, 2005*) or community networks (*Carroll, 2012*), and the brokerage is resulted from establishing social connections, which leads to social capital gain (*Carroll, 2012*; *Burt, 2005*; *Van Liere, 2010*). In contrast curators broker information by creating artifacts in an online social network environment. In this study, we have one participant who obtained a new job in part due to her work on a curation repository. However, whether curation helps curators bargain for better terms in general, and whether curators established social ties and how long such social ties exist as a result of curation requires future investigation.

## Design implications and future directions

Our analysis describes GitHub as a technical infrastructure that meets the needs of curation practices. However, there is still substantial room for curation practices to improve. Our participants found that curating software related contents required a great deal of effort to filter resources and to actively maintain the existing ones. In addition, as the length of

the curated list grows, it also creates navigational difficulties. The current conditions could be improved by (1) empowering curation with automated filtering tools, and (2) adding navigational support within a curated list.

Automated tools can reduce the amount of effort curators need to spend on curating processes. Curators currently do manual selection and evaluation of potential resources as well as eliminating outdated resources. Selections are usually achieved by employing search engines or following recommendations from others. Automated tools can help curators reduce the manual efforts spent on finding and maintaining resource lists. For example, some of our participants manually refer to third party tools to check resources status, such as last-updated-date. An automated tool that checks and filters resources according to query fields can largely diminish the noise and reduce time and efforts to select and evaluate resources. In addition, an automated tool can also help maintain existing resources, by checking whether a software project is still under active development or it is deprecated.

Providing navigational support aims at solving the following issues: (1) lengthy curated list, (2) lacking a search function, and (3) lacking common themes across different curation projects. To be more specific, anchored table of contents, which is fixated on the screen, gives readers a clear structure of a document, as well as enables them to jump among sections. This change would make navigating a long list easier. Adding a search function within a curation repository, allowing users to query keywords of the curated items, can help users explore and find ideal resources promptly. Also, templates can provide common structure and themes in different curation repositories. For instance, our participants mentioned that one of the features they wanted for each curation repository to have was to include a beginner's section, where they could easily find out hands-on resources. Curation repositories could adopt a template that includes commonly identified themes, so that users will be familiar with the structure of different curation repositories and thus locate resources more efficiently.

Future work should seek feedback from GitHub users who are consumers of resources in curation projects. Together with what we have learned from this study, we will design and implement tools to help curators select, evaluate, and maintain resource lists more effectively, and allow users to navigate and retrieve desirable resources readily.

## Limitations

Our study was a qualitative investigation of self-reported curation practices and experiences of GitHub developers. More specifically, we did not carry out a controlled study to manipulate hypothesized causal relationships among constructs. Also, due to our limited sample size, cross tabulations among responses were unlikely to be generalizable to the larger population of curation repository owners. A general limitation of qualitative field methods is that some well-known approaches to validity, associated with positivist science, cannot be employed, such as construct validity, statistical validity, or predictive validity. For a qualitative research design such as ours, credibility and transferability are key validity issues: credibility is whether the results are believable and transferability refers to the degree to which the results can be transferred to other settings (*Guba, 1981*; *Hoepfl, 1997*).

**Credibility**. We tried to ensure credibility by having two researchers code the data independently, and then calculating the kappa statistic to assess coding agreement. In addition, we used member checking in the interview process to have participants directly corroborate findings. These two approaches were encouraging and convergent, indicating very good credibility for our reported findings (*Lincoln & Guba, 1985*). We acknowledge that this is only a starting point in understanding curation practices in GitHub, but it succeeded in raising many issues that could be pursued now in more constrained research designs.

**Transferability**. There are several potential threats to the transferability of this study. First, the owners of the most popular curation repositories did not respond to our interview invitation. Given those celebrity curators' massive audience base and extensively received contributions and attentions, their motivations and practices might be different from the general curators we focused in this study. Second, the way we recruited our participants was to use the keyword ''curated list to search through repository descriptions on GitHub and identified repository owners as our potential interviewers. This search method is transparent and direct, but may have missed curation repositories and owners that did not self-identify with our keywords. Follow-up research can expand the criteria for identifying repositories to further develop our findings. Finally, all of the curators we investigated were recruited on public GitHub, so our results may not generalize to closed source systems. This is another direction for subsequent research to develop our initial findings.

## CONCLUSION

This study seeks to close a gap in the literature by providing a greater understanding of the motivations that software developers appropriate GitHub for curation, and their experiences with that practice. By conducting in-depth interviews with 16 participants about their curation experiences, we uncovered that curators were motivated by altruism, personal needs, and peer recognition, which were comparable to motivations to participate in open source projects. Whether these motivations support long-term participation in curation practice is yet to be discovered.

Curation repository is an appropriation of an online collaborative working tool, indicating that software developers' community starts to leverage GitHub as a tool for communicating socially generated knowledge. It reflects the flexibility and reconfigurability of the tool. And other similar practices, such as sharing course curriculum on GitHub, start to surface.

In addition, curation repositories serve important functions of communities of practice. They support software developers' work, guide learning through an engineering topic, and communications within the community. Curation practice strengthens software developers' community and can help it grow.

Finally, current curation practices are limited by lacking standard formatting, tools for helping curators find and maintain existing curated resources, and reaching the target audience, which creates opportunities for future improvements.

### Funding

The authors received no funding for this work.

### Competing Interests

John M. Carroll is an Academic Editor for PeerJ Computer Science.

### Author Contributions

- Yu Wu conceived and designed the experiments, performed the experiments, analyzed the data, contributed reagents/materials/analysis tools, wrote the paper, prepared figures and/or tables, performed the computation work, reviewed drafts of the paper.
- Na Wang conceived and designed the experiments, analyzed the data, wrote the paper, prepared figures and/or tables, performed the computation work, reviewed drafts of the paper.
- Jessica Kropczynski conceived and designed the experiments, contributed reagents/materials/analysis tools, wrote the paper, reviewed drafts of the paper, suggested relevant works and possible framing.
- John M. Carroll conceived and designed the experiments, contributed reagents/materials/analysis tools, wrote the paper, reviewed drafts of the paper, guided and oversaw the entire process.

### Ethics

The following information was supplied relating to ethical approvals (i.e., approving body and any reference numbers):

The study was approved by Penn State University Institutional Review Board.

### Data Availability

The interview transcripts have been uploaded as Supplemental Files.

### Supplemental Information

Supplemental information for this article can be found online at http://dx.doi.org/10.7717/peerj-cs.134#supplemental-information.

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
