# Peer review of "The appropriation of GitHub for curation"

_PeerJ Computer Science, doi:10.7717/peerj-cs.134_

## Round 0.1 · original submission · Major Revisions

As you see, the reviewers and myself were largely positive about your manuscript. Well done!

However, there are a few text-level changes that I would like to see addressed. The most important ones are summarized below, but please also take the other comments of the reviewers into account:

* Both reviewers commented about your methodology, or at least the description thereof. Please carefully revise and extend how you actually conducted your study, and include rationales.

* Relatedly, I would prefer if you made the threats to the validity of your study a bit more explicit. I understand that the "Limitations" section attempts this, but a more standard way is to have a dedicated section on Threats, discuss them more detailedly, and group them using standard treat classes (e.g., internal threats, external threats, construct threats, ...).

* The discussion and outcome sections are somewhat 'thin' and speculative as it is. Please make sure that what you discuss in the 'Discussion' actually follows or directly relates to the outcomes of your study, as opposed to just be a general reflection of the area and your opinions about it.

·

Basic reporting

* I'd find it helpful if added some examples of resources that are curated (somewhere around page 1 line 40).
* P2,l53: why are personal needs extrinsic?
* There are several smaller errors in the English, e.g. line 211: "participant" -> participate
* Is it possible that you show an overview of your coding schema at the beginning of the results section?

Experimental design

* why do you only do open coding? Why not the further steps of Grounded Theory?
* The relative counting of the occurrences of codes that you use in the results section should be explained here as well.
* I'm also confused by the claim that you use open coding and then your codes are the categories found be other studies on motivation on OS projects. Wasn't it more a preexisting categorisation?

Validity of the findings

* the high kappa indicates a good agreement and hence generalisability of the codes

Additional comments

- While I found the discussion interesting and well-informed, I think it could be better integrated with the results of the interview. Some parts of it feel like they could have written without the interviews.
- The conclusions are very short. Could you please summarise what we should actually conclude? What do we learn and what effect should/could that have on software engineering practice?

Reviewer 2 ·

Basic reporting

PROs:
* The paper is generally well written, and discusses an interesting and timely topic which has not been extensively studied in the past.

CONs:
* At various points in the paper, the authors seem to use language with a certain amount of personal bias or over-emphasis, for example in the statement "[...] provides them with a perfect starting point" on page 8 the word "perfect" might be considered as over-emphasizing. This is not a critical comment, but it does make a subtle difference, and it would be advisable to use cautious and hedging language wherever possible. Also, the authors should carefully review whether all of the presented conclusions are actually drawn from and backed by the survey data collected.

Style/language (minor):
* abstract: "motivates software developers to engaged" -> "motivates software developers to engage"

Experimental design

* The methodology of the paper is primarily based on quoting exemplary study responses for different dimensions/categories of the survey. It is not entirely clear how the authors came up with the dimensions that are discussed in the study. Which of the two came first - 1) were the dimensions defined beforehand and the survey results are then used to discuss/confirm these dimensions, or 2) are the dimensions derived directly from the survey results? Generally, it may be advisable to be a bit more precise and specific about the methodology and how the survey has been constructed. Some details are provided in the supplemental material (e.g., chat messages exchanged with the respondents), but including the actually survey questions in the text (or appendix) would help making the paper more self-contained.

* In addition to the qualitative analysis of the free-text study answers, it would be interesting to see a basic quantitative analysis to provide an overview of the numeric results of the study. Due to the way the results are currently summarized in the paper, the presentation seems fairly lengthy and it is hard for the reader to extract the key insights. Some of the discussed results appear straight-forward or at least not very surprising, but if we had some numbers for these individual results, they could become more meaningful (e.g., X% of respondents gave a similar answer, etc).

Validity of the findings

* I generally wonder if any kind of cross-correlation has been performed. For each of the quoted statements the paper lists the survey respondents (P1-P16) who provided the answer, which in itself does not provide a great added benefit (the paper would be equally valid and valuable if the reference to the respondents were entirely left out). An interesting aspect, though, would be to see how the individual respondents provided answers to different combinations of survey questions. Maybe there is a common theme among the answers? Are any of the survey questions in any way related to one another? (e.g., if a respondent provides answer X to question A, they are likely to provide answer Y to question B) It would be interesting to analyze these aspects more systematically.

* In my view, the Discussion section should be more focused on discussing the results and limitations particularly related to the presented contribution (survey). In its current form, the section seems a bit like an extension of the Background section, rather than a specific discussion and critical reflection of the presented results. This is evidenced by the fact that most of the paragraphs/statements in the Discussion are backed by external resources. Using external references is not a problem in itself, but the paper should draw a clear line between discussion of related work / background, and discussion of the results and limitations of the approach itself.

---

## Round 0.2 · Minor Revisions

I understand and agree with your reluctance to add more quantitative results due to limited sample size. It is my opinion that the experimental design and analysis as presented in this paper is rigorous and adequate, and does not require further revision.

Please address the minor reporting issues that have been raised. Specifically, I agree with the reviewer that there still is a bit of unnecessary redundancy between results and discussion, which could be cleaned up further.

After these steps, I am sure the paper will be good to go.

Reviewer 2 ·

Basic reporting

The authors have made substantial changes in this revised version, some of the most important review comments have been addressed. The readability and structure of the paper has improved from the initial version.

Thanks also for preparing a "diff" version which made it easier to track the changes from the previous version.

A few minor comments:
* line 101: "Software Developers’ Motivations in Participating Online Communities" -> "Software Developers’ Motivations for Participating in Online Communities"
* line 468: "to keep them up-to-date" -> "to keep themselves up-to-date"
* line 502: "a only starting point" -> "only a starting point"
* line 508: "was to used ”curated list”" -> "was to use ”curated list”"

Please give the paper another thorough pass and check for any orthographic issues, typos, etc.

Experimental design

Although the design and methodology have been improved (e.g., by adding the coding scheme in Table 1), I still believe that the paper could have been improved by following a more rigorous methodology for analyzing the data and by presenting the results in a more structured and "condensed" way (i.e., aggregating the results). For example, in addition to quoting exemplary statements by individual respondents, the authors could have put more emphasis on extracting some numbers on differences and commonalities in the respondents' answers. (See also the comments related to "Validity of the findings" below.)

There still seems to be a bit of overlap between the Results and Discussion sections. For example, both sections discuss motivations for curators. That said, the Discussions section has improved from the last version as it now discusses limitations and future directions.

Validity of the findings

I still find it a bit unfortunate that the authors were unable to extract a few more quantitative results in their analysis. Performing a basic numerical analysis and cross-correlation in terms of answers by different respondents could substantially add to the value of the contribution.

It has been mentioned in the rebuttal that the sample size is too small to draw such conclusions. Arguably, this should be mentioned in the paper (Limitations section) as well. Note that, even if the data does not allow for conclusions with a very high level of confidence (e.g., 95+%), the confidence interval could be adjusted (e.g., ~90%) to at least show a general trend which can then be further substantiated with more data in future work. I am hesitant to flag this as a strong requirement for acceptance, but at least I would urge the authors to consider it for the final revision, and for future work.

---

## Round 0.3 · accepted · Accept

After editorial review, I have come to the decision that the paper is now ready for publication. Well done!